# Elevated Expression of Cell Adhesion, Metabolic, and Mucus Secretion Gene Clusters Associated with Tumorigenesis, Metastasis, and Poor Survival in Pancreatic Ductal Adenocarcinoma

**DOI:** 10.3390/cancers16234049

**Published:** 2024-12-03

**Authors:** Karthik Balakrishnan, Yi Xiao, Yuanhong Chen, Jixin Dong

**Affiliations:** Eppley Institute for Research in Cancer and Allied Diseases, Fred & Pamela Buffett Cancer Center, University of Nebraska Medical Center, Omaha, NE 68198, USA; kbalakrishnan@unmc.edu (K.B.); yi.xiao@unmc.edu (Y.X.); cheny@unmc.edu (Y.C.)

**Keywords:** PDAC, differentially expressed genes, gene cluster, tumorigenesis, metastasis, survival

## Abstract

This study investigates differentially expressed genes and their effect on pancreatic ductal adenocarcinoma (PDAC). These differentially expressed genes were identified in PDAC using transcriptomic profiles from the gene expression omnibus repository and the GEO2R web-based tool. Protein–protein interaction analysis was performed using the STRING database to characterize gene clusters. The following three functional gene clusters were identified: (1) extracellular matrix/cell adhesion, (2) metabolism, and (3) mucus secretion. Based on the TCGA database, expression of these gene clusters was associated with PDAC tumorigenesis, metastasis, and patient survival. This study identifies 68 differentially expressed genes that appear to be crucial in PDAC development and prognosis, thus providing potential targets for future therapeutic intervention in pancreatic cancer.

## 1. Introduction

Pancreatic cancer is one of the most prevalent types of tumors, and patients have a minimal chance of long-term survival [1]. By the end of the year 2040, pancreatic cancer is predicted to be the second-leading cause of cancer-related deaths in the United States [2]. Pancreatic ductal adenocarcinoma cancer (PDAC) is the most common form of pancreatic cancer. PDAC is considered therapeutically challenging because 90% of the cases are detected at late stages of cancer progression and are often resistant to chemotherapeutic treatments [3,4]. Although several molecular therapies have been established for PDAC treatment in recent years, the therapeutic benefits remain limited [5,6]. Moreover, the pathophysiologic mechanisms for PDAC initiation and progression remain unclear. The prognosis of PDAC remains poor and correlates to the early malignant spread and aggressive metastases to other organs [7]. The early stages of PDAC cancer are called a “silent” tumor due to nonspecific symptoms and the lack of early prognostic biomarkers [8]. Therefore, identifying a comprehensive gene set to serve as prognostic markers is necessary to help counter the complex mechanisms for PDAC tumorigenesis and metastasis, and help offer an early therapeutic response.

The current study first sought to identify differentially expressed genes in PDAC tumors versus non-tumorous pancreatic tissues by using publicly available mRNA expression profiles from the gene expression omnibus (GEO). Next, the co-expression patterns and functions of the consistently upregulated genes identified (i.e., the PDAC gene set) were further examined in protein–protein interaction (PPI) network constructions. Then, the TCGA database was used to examine the clinicopathological features of these genes. Furthermore, the receiver operating characteristic (ROC) curve, gene set enrichment analysis, and overall survival plot analysis indicated the biological implications and clinical significance of these genes in PDAC patients. Collectively, this study systematically examines the impact of several gene clusters on PDAC tumorigenic features, including primary tumor, tumor stage, grade, lymph node infiltration, and overall survival. As such, the findings herein shed light on the genetic mechanisms underlying the progression of PDAC and can guide strategies for individualized treatments.

## 2. Materials and Methods

### 2.1. Genome-Wide Expression Profiles 

The following datasets comparing expressions in PDAC and non-tumorous samples were collected from the gene expression omnibus (GEO): GSE16515, GSE28735, GSE62165, and GSE43288 [9]. Also used for this study were mRNA expression profiles GSE19280 and GSE42952 with normal pancreatic tissues, PDAC, liver metastasis, and peritoneal metastasis samples. The normalized series matrix files or raw data .CEL files were obtained from GEO. Raw files processed by R software were employed for the MAS5/Robust Multi-array Average algorithm (RMA) normalization with the Affy Package [10,11]. The probes were mapped to unique gene symbols using the relevant platform annotation files. Finally, gene expression values from several gene identification probes were averaged and used in subsequent studies. 

### 2.2. Identification of Differentially Expressed Genes 

The GEO2R tool was used to detect differential gene expression profiles [12,13]. Genes exhibiting up- or downregulated expressions with a fold change ≥2 or <−2 and a *p-*value ≤ 0.05 were considered statistically significant and utilized for further analysis. The Bioinformatics and Evolutionary Genomics Venn diagram (18 September 2024; https://bioinformatics.psb.ugent.be/webtools/Venn/) tool was employed to identify commonly upregulated genes among the expression profiles.

### 2.3. Heatmap Visualizations and Functional Annotations 

The dChip tool was utilized for hierarchical cluster analysis [14]. The hierarchical grouping of samples and genes was displayed using a heatmap representation, with increased expression in red and decreased expression in green. Furthermore, gene set enrichment analysis was used to assess the enrichment scores of the PDAC gene sets [15]. Lastly, molecular signatures databases (MSigDB) were used for ontological functional annotations [16].

### 2.4. Protein–Protein Interaction Network Construction 

The protein–protein interaction was examined using the Search Tool of the Retrieval of Interacting Genes (STRING) database [17]. The study used PDAC gene sets in this database to generate a protein–protein interaction network based on k-means clustering. 

### 2.5. ROC Curve

The ROC curve was employed to examine the specificity and sensitivity of gene clusters. The MedCalc tool was utilized to create the ROC curves [18,19]. 

### 2.6. Survival Curve Analysis

Kaplan–Meier plots were employed to determine the overall survival [20,21]. The Kaplan–Meier plots were combined with information on clinical features and mortality rates to investigate the potential impact of the cluster gene expression on the survival pattern of PDAC cancer patients. The gene expression patterns were analyzed for extracellular matrix/cell adhesion, metabolism, and mucus secretion-related gene clusters. The log-rank test was used to determine the *p* values [22].

### 2.7. Validation in TCGA Profile

The PDAC tumor RNA sequence profile was obtained from the TCGA database using an online portal UALCAN (16 August 2024; https://ualcan.path.uab.edu/analysis.html) [23]. The following genes were used for the analysis of clinicopathological features, such as tumor stages, grade, metastasis, and overall survival: MBOAT2, MGLL, EGLN3, SLC2A1, MUC13, FXYD3, AGR2, TFF1, CDH3, COL17A1, ITGA2, and LAMC2.

### 2.8. Cell Culture and Quantitative Real-Time PCR Analysis

The non-transformed human pancreatic ductal cell line (HPNE) and pancreatic adenocarcinoma cell lines (BxPC3, PANC1, S2.013, and CAPAN2) were cultured at 37 °C with 5% CO_2_. The DMEM/RPMI medium was supplemented with 10% fetal bovine serum, 1x penicillin, and streptomycin. Cell culture, total RNA isolation, reverse transcription, and quantitative real-time PCR were performed as previously described [24,25].

## 3. Results

### 3.1. Identification of Differentially Expressed Genes in PDAC Samples

From the GEO, we collected four mRNA expression profiles (GSE16515, GSE28735, GSE62165, and GSE43288) that comprised PDAC and non-cancerous samples. The GEO2R tool was used to analyze differentially expressed genes within those profiles. A fold change ≥2 or ≤−2, and a *p*-value < 0.05 were used to determine the significantly differentially expressed genes (Appendix A). The expression patterns of upregulated and downregulated genes are shown in the heatmap and volcano plots (Appendix A). These differentially expressed genes were used for further analysis.

Among the differentially expressed genes, the overlapping upregulated genes in three or more profiles were considered PDAC gene sets. These gene sets are shown in the Venn diagram representation (Figure 1A); a total of 68 genes are displayed in Appendix A. The derived PDAC gene sets were found to be expressed more exclusively in the PDAC samples than in non-tumorous tissue (Figure 1B–E). These gene sets were further assessed with gene set enrichment analysis in these four PDAC profiles. The results indicate significant enrichment scores for PDAC tumors (Appendix A). Moreover, gene ontology analysis shows these gene sets were highly involved in matrix complex organization, cell adhesion, and the dysregulation of collagen structural formation in PDAC tumors (Figure 1F).

### 3.2. Protein–Protein Interaction Analysis Identifies Three Gene Clusters Implicated in PDAC: Cell Adhesion, Metabolism, and Mucus Secretion

To investigate co-expression patterns and functional associations of the identified PDAC gene sets, protein–protein interaction analysis was performed using the STRING database. The results show that the PDAC gene sets can be functionally divided into the following three clusters: extracellular matrix/cell adhesion (red), metabolism (blue), and mucous regulation (green) (Figure 2A,B). The first cluster (red) includes the following 40 genes involved in the extracellular matrix/cell adhesion: AHNAK2, ANLN, ANTXR1, CDH11, CDH3, CEACAM5, CEACAM6, COL10A1, COL11A1, COL12A1, COL17A1, COL1A1, COL1A2, COL8A1, CP, CTHRC1, EDIL3, FERMT1, FN1, GJB2, INHBA, ITGA2, ITGA3, KRT7, LAMB3, LAMC2, MMP11, MMP12, NOX4, PCDH7, POSTN, SCEL, SERPINB5, SLPI, SULF1, TCN1, THBS2, TMPRSS4, TRIM31, and VCAN. The second cluster (blue) indicates that EGLN3, MBOAT2, MGLL, and SLC2A1 are associated with metabolic processes in PDAC. The third cluster (green) contains the following 10 genes that are related to mucus secretion in PDAC: AGR2, ANXA10, CLDN18, CTSE, FXYD3, KRT19, MUC13, TFF1, TSPAN1, and VSIG1. The remaining few genes are not included in this study since they do not form clusters or interact with one another. Moreover, genes in these three clusters also display significant upregulation at the mRNA level in PDAC tumors compared to non-cancerous samples (Appendix A). To evaluate whether the expression of these gene clusters displays high sensitivity and specificity in PDAC tumorigenesis, ROC-based analysis was performed. The results suggest the expression patterns of all three gene clusters are sensitive and specific to predict PDAC development, as indicated by significant areas under the curve (AUC) values and *p* values (*p* < 0.05) (Figure 3A–D). Overall, the protein–protein interaction analysis identified three PDAC gene clusters involved in cell adhesion, metabolism, and mucus secretion, and predicted the pathogenesis of PDAC with high sensitivity and specificity.

### 3.3. Increased Expression of Gene Clusters Tied to Shorter Survival and Greater Metastatic Burden in PDAC Patients

To further investigate the clinical relevance of the gene clusters identified, the overall survival rate was first examined. The Kaplan–Meier plots suggest that increased expression of all three clusters correlates to poor survival in PDAC patients (Figure 4A–C). Thus, overexpression of the gene sets appears to indicate poor survival in PDAC patients with significant *p* values.

Next, the impact of these genes on PDAC metastasis was examined. Two GEO profiles (GSE19280 and GSE42952) containing information on normal pancreatic tissues, PDAC, normal liver tissues, PDAC liver metastasis, and/or PDAC peritoneal metastasis were analyzed. Interestingly, the extracellular matrix/cell adhesion gene cluster was found to significantly affect PDAC carcinogenesis, but not PDAC liver metastasis (Figure 5A). Compared to the liver metastasis, peritoneal metastasis was more commonly observed in PDAC patients with high expression of extracellular matrix/cell adhesion genes (Figure 5B). Nevertheless, elevated patterns in liver metastasis (Figure 5C,D) and peritoneal metastasis (Figure 5E,F) were observed in both metabolic and mucus secretion clusters. Therefore, these findings suggest the association between the increased expression of extracellular matrix/cell adhesion-related genes with peritoneal metastasis, whereas overexpression of metabolic and mucus secretion gene clusters is associated with liver and peritoneal metastases in PDAC tumors.

**Figure 2 cancers-16-04049-f002:**
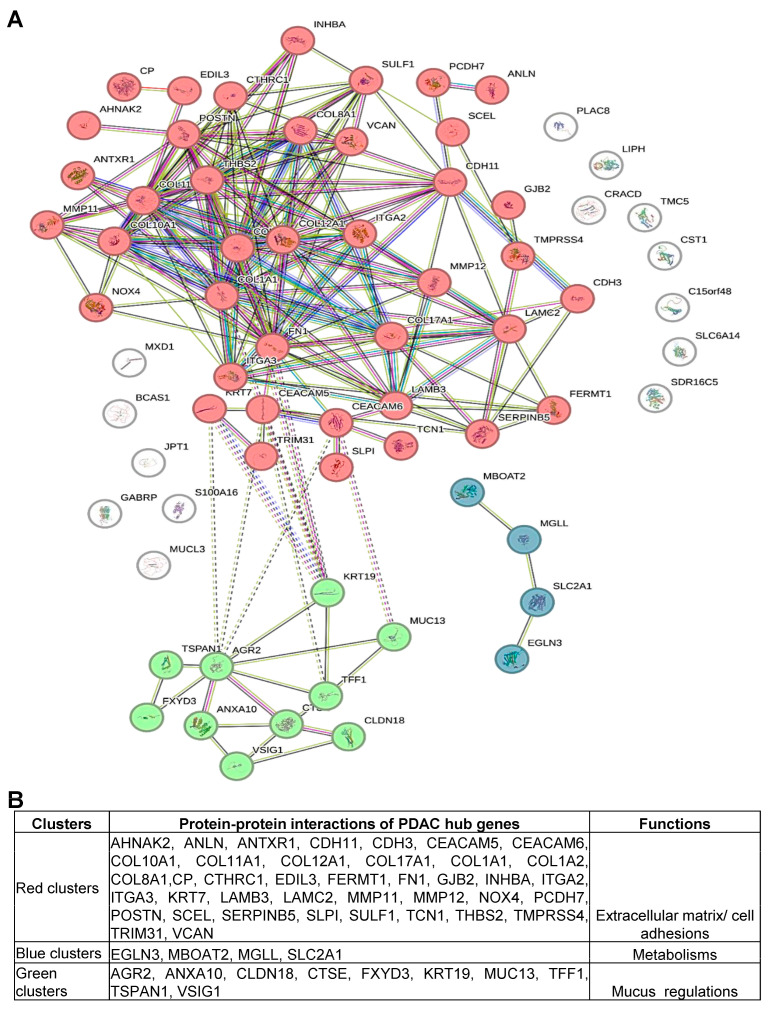
Protein–protein interaction (PPI) analysis identifies three functionally associated gene clusters in the PDAC gene sets. (**A**) The PPI network shows three gene clusters based on k-means clustering for their functional interactions. (**B**) Cluster genes and their biological functions (red cluster: extracellular matrix/cell adhesion; blue cluster: metabolic remodeling; green cluster: mucus secretion).

**Figure 3 cancers-16-04049-f003:**
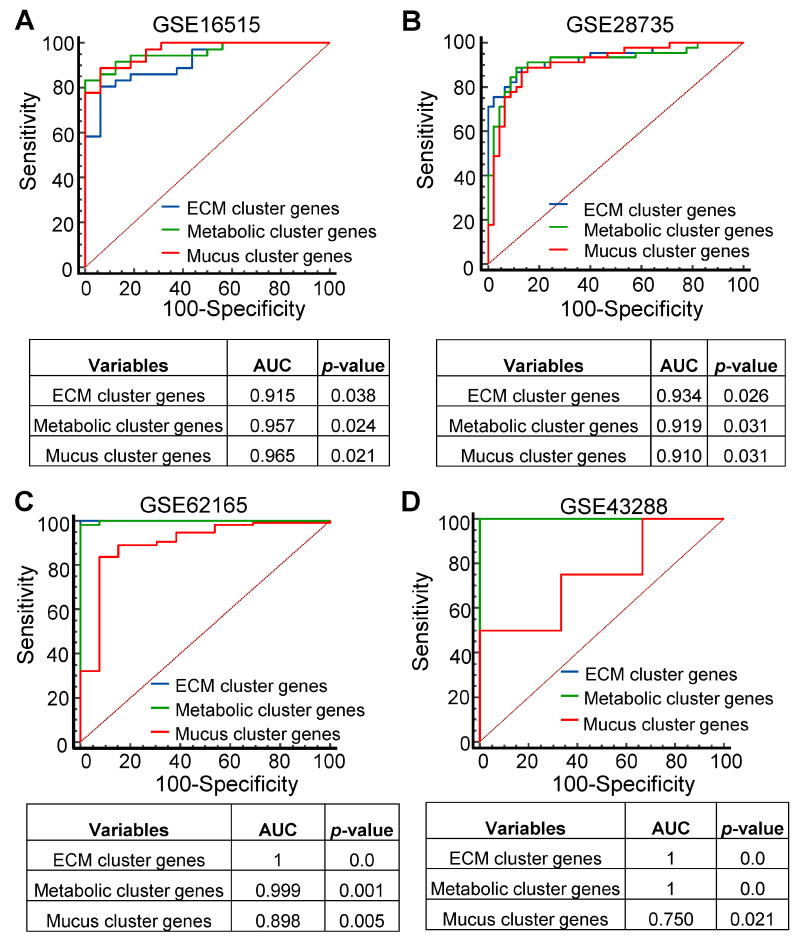
Identified gene clusters predict PDAC pathogenesis with high sensitivity and specificity. (**A**–**D**) Receiver operating characteristic curve shows greater area under the curve (AUC) values with significant *p*-values for three gene clusters in the expression profiles.

**Figure 4 cancers-16-04049-f004:**
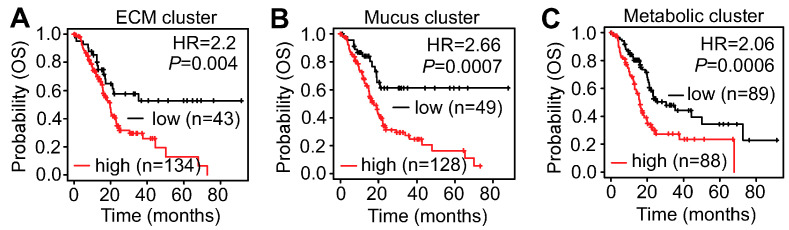
Elevated expression of the identified cluster of genes correlates with the overall survival (OS) rate of PDAC patients. (**A**–**C**) The Kaplan–Meier plots indicate increased expression of all three gene clusters is associated with poor survival in PDAC patients. Data were extracted from the KM plotters database.

**Figure 5 cancers-16-04049-f005:**
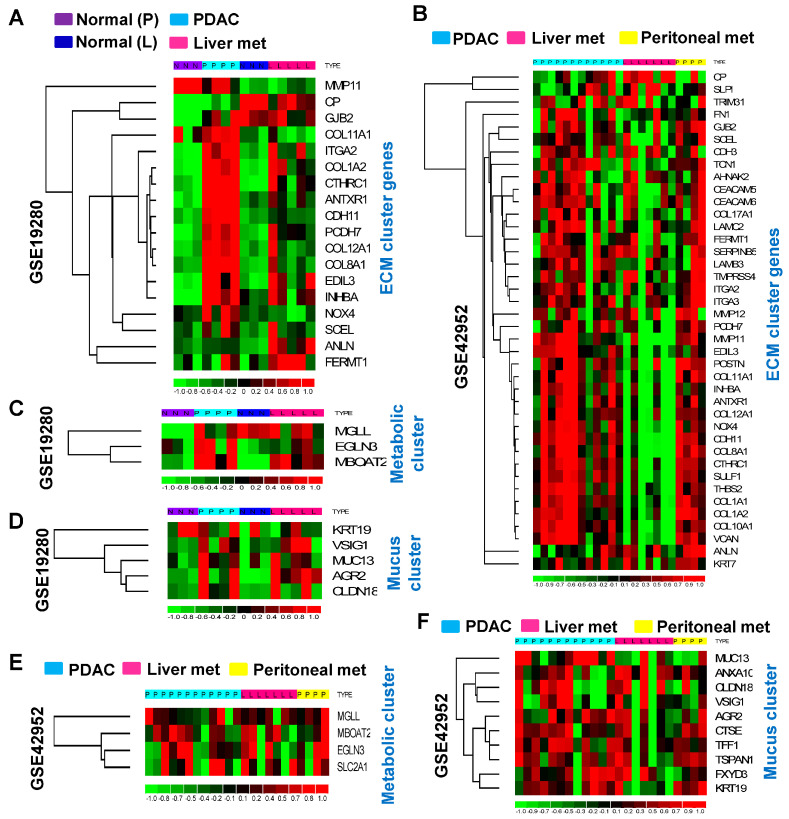
Identified gene clusters are associated with peritoneal or liver metastases in PDAC. The following two datasets were used: GSE19280 and GSE42952 (PDAC, liver, and peritoneal metastasis). (**A**) Extracellular matrix/cell adhesion-associated genes are extensively overexpressed in primary PDAC but not PDAC liver metastasis. (**B**) The expression levels of extracellular matrix/cell adhesion-associated genes are elevated during the PDAC peritoneal metastasis. (**C**–**F**) The expression levels of metabolic and mucus secretion-associated genes are increased in PDAC primary tumors as well as peritoneal and liver metastasis.

### 3.4. Overexpression of the Gene Sets Is Associated with Increased PDAC Tumorigenesis, Cancer Stage, Tumor Grade, Lymph Node Infiltration, and Poor Survival

RNA sequence profiles containing samples like primary PDAC tumors, tumor stages and grades, lymph node infiltrations, and survival outcomes were further extracted from the TCGA database to examine the clinical relevance of gene expression from each cluster. Increased expression of metabolic cluster genes MBOAT2 (Membrane-bound O-Acyltransferase domain containing 2), MGLL (Monoacylglycerol lipase), EGLN3 (Egl-9 family hypoxia-inducible factor 3), and SLC2A1 (Solute carrier family 2 member 1) was observed in primary PDAC tumors compared to non-tumorous tissue (Figure 6A). Increased expression was associated with tumor stages, grades, as well as lymph node infiltration in varying degrees (Figure 6B–D). Elevated expression of MBOAT2, EGLN3, and SLC2A1 correlates with poor PDAC patient survival (Figure 6E). We further validated that their expression levels were upregulated in pancreatic cancer cells when compared with non-cancerous HPNE cells (Figure 6F,G).

The expression of MUC13 (Mucin 13), FXYD3 (FXYD domain-containing ion transport regulator 3), AGR2 (Anterior gradient protein 2), and TFF1 (Trefoil factor 1) was elevated in primary (Figure 7A) and advanced PDAC (Figure 7B,D). AGR2, FXYD3, TFF1, and MUC13 are associated with increased tumor grade. Furthermore, overexpression of FXYD3 correlates with poor survival in PDAC (Figure 7C,E). MRNA levels of AGR2, TFF1, and FXYD3 were greatly increased in pancreatic cancer cells when compared with non-cancerous cells (Figure 7F,G). Surprisingly, MUC13 mRNA was not detected in any of these cell lines. Among the extracellular matrix/cell adhesion gene cluster, CDH3 (Cadherin 3), COL17A1 (Collagen type XVII alpha 1 chain), ITGA2 (Integrin alpha 2), and LAMC2 (Laminin subunit gamma 2) were examined. Similarly, these genes are associated with tumorigenesis, increased tumor stage, increased tumor grade, increased lymph node infiltration, and poor survival in PDAC patients (Appendix A). Together, these data suggest that the elevated expression of genes in the cell adhesion, metabolic, and mucus secretion clusters can help predict tumorigenesis, cancer progression, and poor patient outcomes in PDAC tumors.

## 4. Discussion

PDAC is a common digestive organ cancer with a poor clinical outcome [5]. Moreover, the incidence rates and death rates of PDAC are still rising globally [26,27]. The clinical features and mechanisms of PDAC tumors remain unclear, and the prognostic factors remain poorly understood [28]. In recent years, for numerous types of cancers, microarrays, and RNA sequencing-based genome-wide expression profile analyses were used to identify biomarkers to help better predict the tumor location, progression, and prognosis [29]. 

Using existing databases and bioinformatic tools, this study identified 68 genes that are commonly upregulated in three or more PDAC profiles. These genes were clustered into three groups: the extracellular matrix/cell adhesion gene cluster, the metabolic gene cluster, and the mucus secretion-related gene cluster. The extracellular matrix/cell adhesion group includes the following genes: AHNAK2, ANLN, ANTXR1, CDH11, CDH3, CEACAM5, CEACAM6, COL10A1, COL11A1, COL12A1, COL17A1, COL1A1, COL1A2, COL8A1, CP, CTHRC1, EDIL3, FERMT1, FN1, GJB2, INHBA, ITGA2, ITGA3, KRT7, LAMB3, LAMC2, MMP11, MMP12, NOX4, PCDH7, POSTN, SCEL, SERPINB5, SLPI, SULF1, TCN1, THBS2, TMPRSS4, TRIM31, and VCAN. The metabolic cluster includes the following genes: EGLN3, MBOAT2, MGLL, and SLC2A1. The mucus secretion group contains the following genes: AGR2, ANXA10, CLDN18, CTSE, FXYD3, KRT19, MUC13, TFF1, TSPAN1, and VSIG1. Our findings suggest that these gene clusters are associated with PDAC pathogenesis such as primary tumors, stage, grade, metastasis, and poor survival. 

In line with our observations, previous studies have shown that many of these genes play critical roles in various cancers, including PDAC. For example, overexpression of EGLN3 through the transforming growth factor beta 1 (TGF ß1) pathway has been found to be linked to cancer cell invasion, metastasis, and poor prognosis in pancreatic cancer [30]. In numerous tumors, EGLN3 was implicated in cancer cell invasion, migration, and proliferation [31]. EGLN3 is a crucial regulator of epithelial–mesenchymal transition (EMT), metastasis, and therapeutic resistance in the lung tumor microenvironment. It also connects the control of EMT and hypoxia signaling by an oxygen sensor [32]. Likewise, MBOAT2 has been shown to be noticeably elevated in pancreatic tumors and was linked to EMT, cancer cell proliferation, migration, grade, recurrence, and poor prognosis [33,34]. In lung cancer, the upregulation of MGLL was found to accelerate tumor progression and metastases, and resultantly, lead to a poor prognosis [35]. Interestingly, MGLL was also associated with aberrant lipid metabolism during lung tumor invasion and metastasis [36]. In colorectal adenocarcinoma, SLC2A1(also called Glut1) expression has been associated with poor survival, tumor grade, stage, metastasis [37], reduced immune cell activity, and poor prognosis [38]. Furthermore, SLC2A1 has been linked to a poor response to adjuvant immunochemotherapy [39]. In PDAC, glucose absorption and lactate release are notable metabolic adaptations of EMT implicated in angiogenesis, metastasis, energy metabolism, chemoresistance, and proliferation [40,41].

A connection between our cluster of mucus genes (AGR2, ANXA10, CLDN18, FXYD3, MUC13, TFF1, and VSIG1) and carcinogenic features has been reported in various recent studies. For example, the secreted protein AGR2 is primarily engaged in cell invasion, motility, drug sensitivity, metastasis, and growth of PANC-1 pancreatic cancer cells [42] and the invasion of colorectal cancer cells [43]. AGR2 plays a crucial role in the regulation of EMT. AGR2 knockout inhibited TGF-β-induced EMT in lung adenocarcinoma [44]. It resulted in alterations of focal adhesion, ECM interaction, and actin cytoskeleton regulation that affect cell EMT induction, migration, and invasion in PDAC tumors [45,46]. Moreover, ANXA10 was linked to the EMT processes in oral squamous cell carcinoma [47], facilitating the migration of melanoma cancer [48], and stimulating the EMT during liver cancer metastasis [49]. CLDN18 expression was positively correlated with the metastatic parthenogenesis of the diffuse gastric cancer subtype and an independent prognostic factor for gastric cancer patients [50,51]. Upregulated FXYD3 expression appears to increase angiogenesis in hepatocellular carcinoma tumors [52] and promotes oxaliplatin resistance in human colorectal cancer [53]. The high expression of FXYD3 was linked to a bad prognosis for patients with pancreatic cancer and stimulated the growth, invasion, and migration of pancreatic cancer cells [54]. FXYD3 could be a biomarker for early-stage pancreatic cancer [55]. Evidence suggests that MUC13 acts as an oncogenic glycoprotein and regulates the progression of esophageal and lung cancer [56,57]. The overexpression of MUC13 can cause abnormal cellular polarity, which increases the adaptability of cancer. This enables chemoresistance, metastasis, and the EMT [58], and it rewires abnormal glucose metabolism to increase the aggressiveness of pancreatic cancer [59]. The upregulation of TFF was found to stimulate the malignant behavior of colon cancer by triggering the EMT process [60] and can potentially serve as a biomarker for the early recurrence of colorectal cancer [61]. Interestingly, the EMT phenotype may develop in pancreatic cancer cells with reduced TFF1 expression. On the other hand, because of the MET process, elevated TFF1 expression autonomously invades and maintains the ability to cause metastatic lesions in distant organs. Therefore, TFF1 expressions can function to suppress EMT (invasion) and encourage MET (metastasis) [62]. VSIG1 also promoted the EMT of gastric cancer [63]. Together, these studies further support the hypothesis that these mucus genes are implicated in features of pancreatic tumor carcinogenic and metastases.

Genes involved in extracellular matrix remodeling and cell adhesion play important functions in various aspects of cancer development. For example, increased expression of the CDH3 gene promotes proliferation, migration, invasion, and chemoresistance in oral squamous cell carcinoma, lung cancer, and thyroid cancer [64,65,66]. The overexpression of COL11A1 and COL1A1 plays a crucial role in the metastasis of gastric, breast, and colorectal cancers, as well as in the development of tamoxifen resistance in breast cancers [67,68]. Furthermore, COL17A1 stimulates the EMT, migration, invasion, and proliferation of pancreatic cancer [69] and is used as a diagnosis biomarker in both pancreatic cancer and lung cancer [70,71]. KRT19 is a promising biomarker for diagnosis and prognosis in ovarian cancer [72] and may contribute to the onset and progression of breast cancer [73]. ITGA2 overexpression facilitates the growth of ovarian cancer, confers paclitaxel resistance [74], and is associated with poor prognosis in PDAC tumors [75]. Elevated expression of the ITGA2 protein serves as a mechanical indicator of matrix stiffness and may lead to gemcitabine chemoresistance [76]. LAMC2 modulates the EMT and increases gemcitabine resistance in PDAC [77]. LAMC2 also correlates with decreased survival, metastasis, and advanced tumor stages in pancreatic cancer [78]. Recent studies showed that the increased expression of SULF1 increases metastasis and cisplatin resistance in gastric cancer [79]. Furthermore, SULF1 is associated with poor prognosis in the case of breast cancer [80]. Taken together, these studies reaffirm our bioinformatics-based analysis and suggest that these extracellular matrix genes (CDH3, COL11A1, COL1A1, COL17A1, KRT19, ITGA2, LAMC2, and SULF1) are highly associated with PDAC tumor carcinogenesis and peritoneal metastasis. Thus, we identified three types of functional clustered genes and explored their carcinogenic feature activities of PDAC tumors in the study.

Additionally, we validate how these clustered genes affected clinicopathological features in pancreatic ductal adenocarcinoma using the TCGA database. The present results revealed three clusters of genes implicated in PDAC clinicopathological features like tumor stage, grade, metastasis, and overall survival. Thus, our study identified PDAC-specific prognostic marker genes for target treatments. However, we must mention certain restraints of our study. Due to the expression profiles being obtained from public databases (GEO and TCGA), it was difficult to evaluate dataset quality. Therefore, conclusions drawn from this study will be further examined. 

## 5. Conclusions

The present study identifies PDAC gene sets from differentially expressed genes of PDAC tumors. Gene set enrichment, gene ontology, protein–protein interaction, ROC curve, and survival analyses further characterized the PDAC gene set expression patterns. Cell adhesion, metabolic, and mucus secretion gene clusters are found to be highly involved in primary and metastatic tumors, as well as survival outcomes in PDAC tumorigenesis. The identified gene clusters have a strong potential as PDAC diagnosis and prognosis biomarkers and to inform future therapeutic targets for treating pancreatic cancer. 

## Figures and Tables

**Figure 1 cancers-16-04049-f001:**
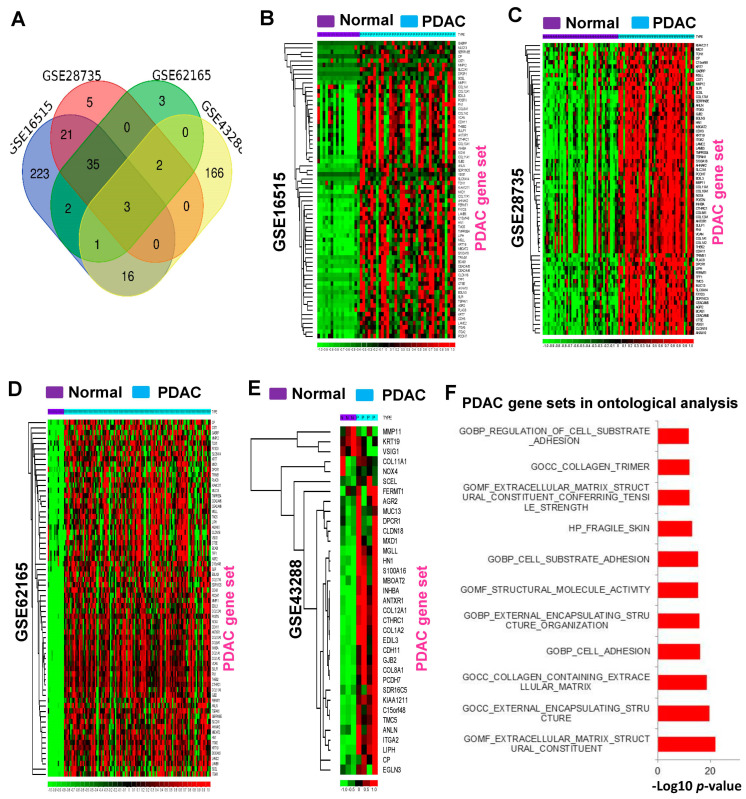
The expression patterns of commonly upregulated genes in PDAC tumors. (**A**) The Venn diagrammatic representation of upregulated genes in PDAC. (**B**–**E**) The expression levels of identified PDAC gene sets are significantly elevated in PDAC tumors. (**F**) Ontological analysis of PDAC gene sets. Extracellular matrix complex organization, cell adhesion, and collagen structural formation were identified as significantly altered biological processes during the carcinogenesis of PDAC tumors.

**Figure 6 cancers-16-04049-f006:**
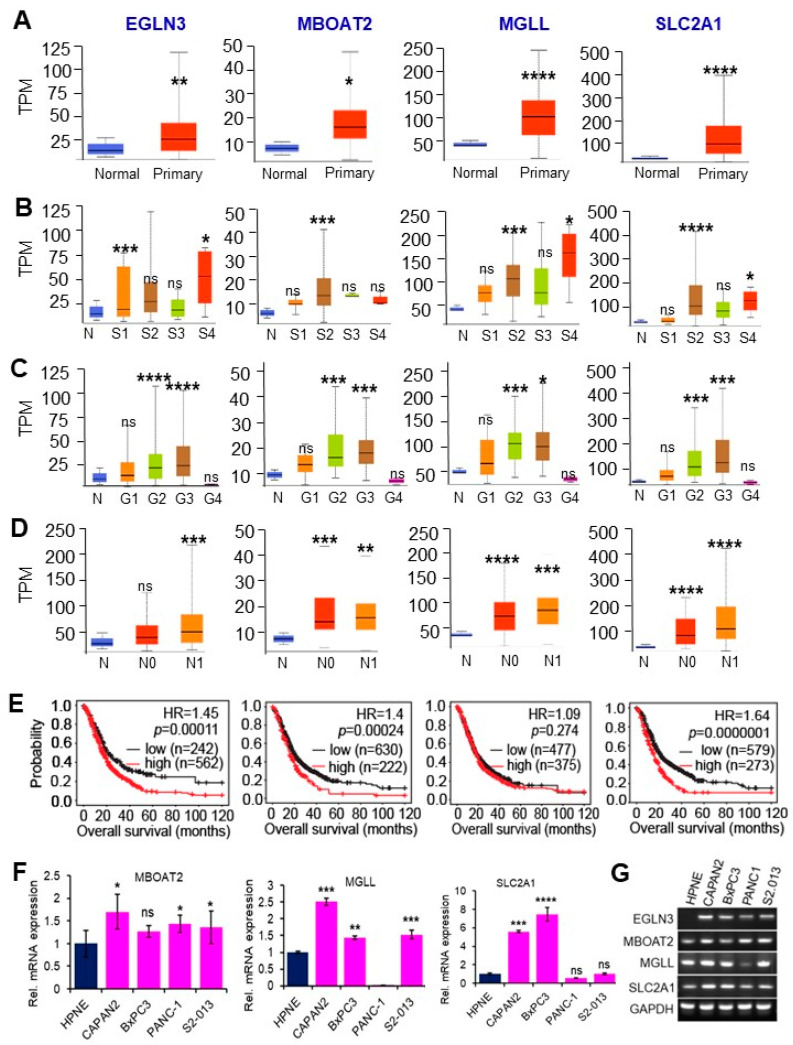
Clinical association of the metabolic genes MBOAT2, MGLL, EGLN3, and SLC2A1 in PDAC tumorigenesis, progression, and patient survival. (**A**) The expression levels of MBOAT2, MGLL, EGLN3, and SLC2A1 are elevated in PDAC tumors compared to non-cancerous tissues. (**B**,**C**) Elevated expression of MBOAT2, MGLL, EGLN3, and SLC2A1 is observed in advanced stages (**B**) and high-grade PDAC tumors (**C**). (**D**) Elevated expression of MBOAT2, MGLL, EGLN3, and SLC2A1 is associated with lymph node infiltration in PDAC. (**E**) Overexpression of MBOAT2, EGLN3, and SLC2A1 correlates with poor PDAC patient survival. Data were extracted from the TCGA database. TPM: transcripts per million; N: normal, N0: node 0, N1: node 1; G1–G4: grade 1 to grade 4; S1–S4: stage 1 to stage 4. ns: not significant. (**F**,**G**) Expression validation by RT-PCR in pancreatic non-cancerous (HPNE) and cancer cell lines (CAPAN2, BxPC3, and PANC1, S2.013). EGLN3 mRNA was not detectable in HPNE (**G**), thus the relative mRNA expression could not be plotted. ****: *p* < 0.0001, ***: *p* < 0.001, **: *p* < 0.01, *: *p* < 0.05 (Student’s *t*-test in (**A**–**D**,**F**); log-rank test in (**E**)). ns: not significant. The uncropped original gel images are shown in Appendix A.

**Figure 7 cancers-16-04049-f007:**
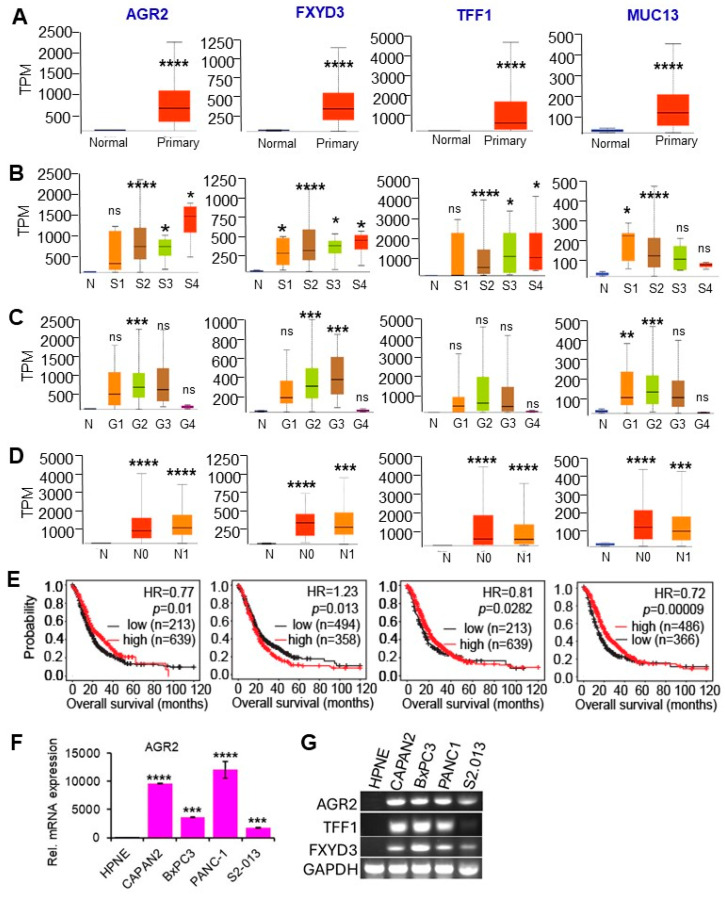
Clinical association of the mucus secretion genes MUC13, FXYD3, AGR2, and TFF1 in PDAC tumorigenesis, progression, and survival. (**A**,**B**) The expression levels of MUC13, FXYD3, AGR2, and TFF1 are elevated in the primary (**A**) and advanced stages (**B**) of PDAC. (**C**) Elevated expressions of AGR2, FXYD3, and MUC13 are associated with increased tumor grade. (**D**) Increased expression levels of AGR2, FXYD3, and MUC13 are associated with lymph node infiltration in PDAC. (**E**) Overexpression of FXYD3 correlates with poor survival. TPM: transcripts per million; N: normal, N0: node 0, N1: node 1; G1–G4: grade 1 to grade 4; S1–S4: stage 1 to stage 4. ns: not significant. (**F**,**G**) Expression validation by RT-PCR in pancreatic non-cancerous (HPNE) and cancer cell lines (CAPAN2, BxPC3, and PANC1, S2.013). TFF1 and FXYD3 mRNAs were not detectable in HPNE (**G**), thus their relative mRNA expression could not be plotted ****: *p* < 0.0001, ***: *p* < 0.001, **: *p* < 0.01, *: *p* < 0.05 (Student’s *t*-test in (**A**–**D**,**F**); Log-rank test in (**E**)). MUC13 mRNA was not detectable in all cell lines. The uncropped original gel images are shown in Appendix A.

## Data Availability

The original data presented in the study are in a publicly accessible repository, and the data presented in this study are available on request from the corresponding author for valuable reasons.

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
