# Peer review of "Elevated Expression of Cell Adhesion, Metabolic, and Mucus Secretion Gene Clusters Associated with Tumorigenesis, Metastasis, and Poor Survival in Pancreatic Ductal Adenocarcinoma"

_cancers, 2024, doi:10.3390/cancers16234049_

Round 1

Reviewer 1 Report (Previous Reviewer 1)

Comments and Suggestions for Authors

The authors have significantly improved the paper. Therefore, I recommend the revised version for publication in cancers.

Reviewer 2 Report (Previous Reviewer 2)

Comments and Suggestions for Authors

I have no further questions regarding this version.

This manuscript is a resubmission of an earlier submission. The following is a list of the peer review reports and author responses from that submission.

Round 1

Reviewer 1 Report

Comments and Suggestions for Authors

Manuscript ID: cancers-3255576 “Elevated expression of cell adhesion, metabolic, and mucus secretion gene clusters associated with tumorigenesis, metastasis, and poor survival in pancreatic ductal adenocarcinoma”

In this paper, the authors explored the expression level of a series of genes present in pancreatic ductal adenocarcinoma (PDAC) tumors. The authors concluded that the specific set of gene expression levels in PDAC tumors along with cell adhesion to the extracellular matrix (ECM), metabolism, and mucus secretion of the gene clusters are the main factors for diagnosis of primary stage and metastatic stage of tumors. They showed that the specified gene clusters can be the best biomarkers for PDAC diagnosis. The subject is interesting and is suitable for the Cancers. However, I would recommend the authors to incorporate the following comments before this paper can be considered for publication in Cancers.

I strongly suggest the authors to consider multiple techniques alongside the immunohistochemistry to verify their findings:

• The quantitative reverse transcription polymerase chain reaction (qRT-PCR) by measuring the gene expression levels of the PDAC tumors through quantification of their mRNA levels in the cells of the tissues. Another technique which needs to be considered for this study is the enzyme linked immunosorbent assay (ELISA).

These two techniques will provide very precise quantitative measurements of the genes of the interest in the tissue samples. The immunohistochemistry also provides a detailed information regarding the levels of expressions of the specific target proteins and therefore it might be a complementary technique to rely for making such significant conclusion as it was made by the authors for this study

Author Response

Responses to the Reviewer’s comments:

Reviewer-1

In this paper, the authors explored the expression level of a series of genes present in pancreatic ductal adenocarcinoma (PDAC) tumors. The authors concluded that the specific set of gene expression levels in PDAC tumors along with cell adhesion to the extracellular matrix (ECM), metabolism, and mucus secretion of the gene clusters are the main factors for diagnosis of primary stage and metastatic stage of tumors. They showed that the specified gene clusters can be the best biomarkers for PDAC diagnosis. The subject is interesting and is suitable for the Cancers. However, I would recommend the authors to incorporate the following comments before this paper can be considered for publication in Cancers.

Query-1: I strongly suggest the authors to consider multiple techniques alongside the immunohistochemistry to verify their findings: The quantitative reverse transcription polymerase chain reaction (qRT-PCR) by measuring the gene expression levels of the PDAC tumors through quantification of their mRNA levels in the cells of the tissues. Another technique which needs to be considered for this study is the enzyme linked immunosorbent assay (ELISA). These two techniques will provide very precise quantitative measurements of the genes of the interest in the tissue samples. The immunohistochemistry also provides a detailed information regarding the levels of expressions of the specific target proteins and therefore it might be a complementary technique to rely for making such significant conclusion as it was made by the authors for this study.

Response:

Thanks for the suggestions. We have performed RT-PCR in non-cancer (HPNE) and cancer cell lines (BxPC3, CAPAN2, S2.013, PANC1). We were able to validate the expression levels of most of these genes (except for MUC13, whose mRNA was not detectable in any of these cell lines we used). Data were added to Figures 6 (Fig. 6F, G) and 7 (Fig. 7F, G).

We shall look forward to hearing from you at your earliest convenience.

Sincerely Yours,

Jixin Dong, Ph.D.

Professor, Eppley Institute for Research in Cancer and Allied Diseases, Fred & Pamela Buffett Cancer Center, University of Nebraska Medical Center, Omaha, NE, 68198, USA. Email: dongj@unmc.edu

Reviewer 2 Report

Comments and Suggestions for Authors

In this study, the authors identified differentially expressed genes (DEGs) in pancreatic ductal adenocarcinoma (PDAC) that are associated with tumorigenesis, metastasis, and patient survival. The authors used transcriptomic data from the Gene Expression Omnibus (GEO) and the GEO2R tool to analyze the normal tissue and PDAC. By using Molecular Signatures Database (MSigDB), protein interaction, and the STRING database, the authors found that there were 68 upregulated genes involved in extracellular matrix, cell adhesion, metabolism, and mucus secretion. This study uncovered the critical role these gene sets play in PDAC development and their potential as diagnostic and prognostic biomarkers for treating pancreatic cancer. Comments below:

Major:

The data regarding mucus cluster genes were generated from the public database; please discuss these mucus gene clusters in the pancreatic exocrine system. 

The results presented the associations between extracellular matrix/cell adhesion gene overexpression and peritoneal metastasis, as well as metabolic and mucus secretion genes with liver and peritoneal metastasis. Nonetheless, the potential mechanistic detail and experimental validation to substantiate these findings somehow still need to be further investigated or discussed. Please discuss these findings on the potential contribution to metastasis mechanisms such as epithelial-mesenchymal transition (EMT), ECM degradation, metabolic reprogramming, immune evasion, enhanced cell survival, chemoresistance, etc.

Minor:

Figures 6 and 7, please add the full name of TPM and explain the method of the statistical analysis. Please add the definitions of S, G, and N.

Author Response

Responses to the Reviewer’s comments:

Reviewer-2

In this study, the authors identified differentially expressed genes (DEGs) in pancreatic ductal adenocarcinoma (PDAC) that are associated with tumorigenesis, metastasis, and patient survival. The authors used transcriptomic data from the Gene Expression Omnibus (GEO) and the GEO2R tool to analyze the normal tissue and PDAC. By using Molecular Signatures Database (MSigDB), protein interaction, and the STRING database, the authors found that there were 68 upregulated genes involved in extracellular matrix, cell adhesion, metabolism, and mucus secretion. This study uncovered the critical role these gene sets play in PDAC development and their potential as diagnostic and prognostic biomarkers for treating pancreatic cancer. Comments below:

 Major Query-1: The data regarding mucus cluster genes were generated from the public database; please discuss these mucus gene clusters in the pancreatic exocrine system. The results presented the associations between extracellular matrix/cell adhesion gene overexpression and peritoneal metastasis, as well as metabolic and mucus secretion genes with liver and peritoneal metastasis. Nonetheless, the potential mechanistic detail and experimental validation to substantiate these findings somehow still need to be further investigated or discussed. Please discuss these findings on the potential contribution to metastasis mechanisms such as epithelial-mesenchymal transition (EMT), ECM degradation, metabolic reprogramming, immune evasion, enhanced cell survival, chemoresistance, etc.

Response: Thanks for the suggestions. We have added discussion parts in the revised Manuscript.

Minor Query-1: Figures 6 and 7, please add the full name of TPM and explain the method of the statistical analysis. Please add the definitions of S, G, and N.

Response: Per your suggestions, we have added these definitions in the revised manuscript.

We shall look forward to hearing from you at your earliest convenience.

Sincerely Yours,

Jixin Dong, Ph.D.

Professor, Eppley Institute for Research in Cancer and Allied Diseases, Fred & Pamela Buffett Cancer Center, University of Nebraska Medical Center, Omaha, NE, 68198, USA. Email: dongj@unmc.edu